# Nrf2 Downregulation Contributes to Epithelial-to-Mesenchymal Transition in *Helicobacter pylori*-Infected Cells

**DOI:** 10.3390/cancers14174316

**Published:** 2022-09-02

**Authors:** Sarah Bacon, Lornella Seeneevassen, Alison Fratacci, Faustine Rose, Camille Tiffon, Elodie Sifré, Maria M. Haykal, Maya M. Moubarak, Astrid Ducournau, Lucie Bruhl, Stéphane Claverol, Caroline Tokarski, Alina-Roxani Gouloumi, Ioannis S. Pateras, Thomas Daubon, Philippe Lehours, Christine Varon, Océane C. B. Martin

**Affiliations:** 1INSERM U1312 BRIC BoRdeaux Institute of onCology, Université de Bordeaux, 33077 Bordeaux, France; 2Institut Gustave Roussy, Université Paris-Saclay, Inserm U981, Biomarqueurs Prédictifs et Nouvelles Stratégies Thérapeutiques en Oncologie, 94800 Villejuif, France; 3Centre National de la Recherche Scientifique (CNRS), Institut de Biochimie et Génétique Cellulaires (IBGC), Unité Mixte de Recherche 5095, Université de Bordeaux, 33077 Bordeaux, France; 4Centre National de Référence des Campylobacters et Helicobacters, CHU de Bordeaux, 33077 Bordeaux, France; 5Plateforme Proteome, University Bordeaux, F-33000 Bordeaux, France; 6Centre National de la Recherche Scientifique (CNRS), Bordeaux Institut National Polytechnique (INP), Institute of Chemistry & Biology of Membranes & Nano-objects (CBMN), Université de Bordeaux, Unité Mixte de Recherche 5248, F-33600 Pessac, France; 72nd Department of Pathology, “Attikon” University Hospital, Medical School, National and Kapodistrian, University of Athens, 104 31 Athens, Greece

**Keywords:** gastric cancer, *Helicobacter pylori*, Nrf2, redox homeostasis, epithelial-to-mesenchymal transition

## Abstract

**Simple Summary:**

Gastric cancer is mainly linked to *Helicobacter pylori* infection. It is therefore important to decipher the mechanisms involved in *H. pylori*-induced gastric carcinogenesis, and especially the early events. We have previously demonstrated that the infection leads to an epithelial-to-mesenchymal transition (EMT) favoring gastric carcinogenesis. *H. pylori* infection is also associated with high levels of oxidative stress. In this work, we aimed at investigating the modulation of Nrf2, a major regulator of cellular antioxidant response to oxidative stress, upon infection with *H. pylori* and to decipher its implication in EMT. We demonstrated that *H. pylori*-induced Nrf2 downregulation may participate in gastric cells’ EMT, one crucial tumorigenic event in gastric cancer. These results could pave the way for new therapeutic strategies using Nrf2 modulators to reduce gastric carcinogenesis associated with *H. pylori* infection.

**Abstract:**

Background: Gastric cancer, the fifth most common cancer worldwide, is mainly linked to *Helicobacter pylori* infection. *H. pylori* induces chronic inflammation of the gastric mucosa associated with high oxidative stress. Our study aimed at assessing the implication of Nrf2, a major regulator of cellular redox homeostasis, in *H. pylori*-induced gastric carcinogenesis. Methods: Using three different gastric epithelial cell lines, a non-cancerous (HFE-145) and two different subtypes of gastric cancer (AGS and MKN74), we analyzed the modulation of Nrf2 expression over time. After invalidation of Nrf2 by CRISPR-cas9, we assessed its role in *H. pylori*-induced epithelial-to-mesenchymal transition (EMT). Finally, we evaluated the expression of Nrf2 and ZEB1, a central EMT transcription factor, in human gastric tissues. Results: We first demonstrated that the Nrf2 signaling pathway is differentially regulated depending on the infection stage. Rapidly and transiently activated, Nrf2 was downregulated 24 h post-infection in a VacA-dependent manner. We then demonstrated that Nrf2 invalidation leads to increased EMT, which is even exacerbated after *H. pylori* infection. Finally, Nrf2 expression tended to decrease in human patients’ gastric mucosa infected with *H. pylori*. Conclusions: Our work supports the hypothesis that Nrf2 downregulation upon *H. pylori* infection participates in EMT, one of the most important events in gastric carcinogenesis.

## 1. Introduction

Gastric cancer (GC) is currently the fifth most common and the fourth deadliest cancer worldwide [1]. GC are mainly gastric adenocarcinomas which incidence, bad prognosis, cellular and molecular heterogeneity make this disease a major health issue. A better understanding of the earliest events leading to gastric carcinogenesis is thus a necessity. Two main common histological subtypes are described according to the Laurén classification: the intestinal type and the diffuse type GC [2].

Different risk factors are known to affect GC incidence, among which *Helicobacter pylori* (*H. pylori*) infection is the essential one. *H. pylori* infection involves almost 50% of the world’s population and induces chronic inflammation of the gastric mucosa, 5 to 15% of which evolve in gastric and duodenal ulcers and less than 1% in GC [2]. Curing this infection remains challenging for clinicians, mainly because of bacterial resistance towards the few available antibiotics [3]. *H. pylori* is a mobile Gram-negative bacterium possessing an urease activity and different virulence factors [4]. Among them, the cytotoxin-associated gene product (CagA) and the vacuolating cytotoxin (VacA) have been described to both exhibits cytotoxic and immunomodulatory activities. CagA, the most studied *H. pylori* virulence factor, which is described as an oncoprotein, is encoded by the cag pathogenicity island (cagPAI) and is injected into the cell through a Type 4 secretion system (T4SS). In the cell, it induces cellular alterations that impair cell motility, cellular proliferation, and apoptosis and causes cytoskeleton rearrangements [5]. The VacA protein, when released by *H. pylori*, is internalized in the host cells, where it can lead to multiple pathogenic effects, such as vacuolation cytotoxicity and apoptosis. It can also result in disruption of endocytic trafficking, mitochondrial perturbations, depolarization of membrane potential and efflux of various ions, and activation of MAP kinases [4]. Clinical strains are often cagPAI+ and possess an active VacA protein, and both factors have been associated with GC increased risk [5,6].

The epithelial-to-mesenchymal transition (EMT), a program that converts epithelial cells to cells with mesenchymal features, plays an important role in both cancer development and progression and is orchestrated by transcription factors (EMT-TFs) [7]. Many studies have demonstrated that infection with *H. pylori* induces the EMT program [8,9,10,11,12]. More especially, infection with CagA+ *H. pylori* leads to the acquisition of the so-called “hummingbird phenotype”, characterized by an elongated cell morphology commonly observed in EMT. Infected cells also present significantly higher expression of the EMT-TFs ZEB1 and Snail [8,9,11].

Another key feature of GC is the exacerbation of oxidative stress linked to *H. pylori* infection. Indeed, the infection induces an important production of reactive oxygen species (ROS), mediated by both virulence factors, CagA and VacA. The accumulated ROS cause DNA damage and subsequently increase genomic instability, providing a pathological basis for the development of gastric carcinogenesis [13,14]. The importance of oxidative stress in GC has been largely described [15].

Nrf2, Nuclear factor (erythroid-derived 2)-like 2, is a transcription factor that plays a critical role in maintaining cellular redox homeostasis by activating the transcription of numerous genes (Heme oxygenase-1, HO1; NAD(P)H quinone dehydrogenase 1, NQO1; and both subunits forming glutathione GSH: glutamate-cysteine ligase modifier, GCLM and glutamate-cysteine ligase catalytic, GCLC) involved in the antioxidant response [16]. In the case of ROS accumulation, Nrf2 translocates into the nuclei, where it interacts with some protein factors, including small Maf (sMaf), and binds to the Antioxidant Responsive Element (ARE), leading to increased transcription of antioxidant genes to restore redox homeostasis [17]. The role of Nrf2 in the protection against oxidative stress in the gastrointestinal tract has been reviewed [18]. In addition, the effect of Nrf2 on EMT transition has been studied in different pathologies, such as pulmonary fibrosis [19,20] and cancer [21,22,23], but has never been described in the context of *H. pylori* infection.

Despite the importance of *H. pylori*-induced oxidative stress in GC, only a few studies have investigated the transcription factor Nrf2 in the context of the infection. We therefore aimed at evaluating the modulation of Nrf2 upon infection with the CagA+VacA+ *H. pylori* strain, using three different epithelial gastric cell lines, a non-cancerous (HFE-145), a diffuse subtype of GC (AGS) and an intestinal subtype of GC (MKN74). We then invalidated Nrf2 in the three cell lines by CRISPR-cas9 to assess its implication in EMT, notably through proteomic analysis. Finally, we assessed Nrf2 levels in *H. pylori*-infected human gastric tissues samples and using the UALCAN database. Moreover, we evaluated in those tissues the correlation between Nrf2 and the EMT-TF ZEB1 as well as in the overall survival probability of GC patients using the Kmplot molecular database.

## 2. Materials and Methods

### 2.1. Gastric Epithelial Cell Lines’ Culture

HFE-145 (non-tumorigenic, immortalized, human gastric epithelial cell line, a kind gift from D. Smoot (Howard University, Washington, DC, USA) [24]) and AGS (human diffuse type of GC, ATCC CRL-1739) were cultivated in Dulbecco’s modified eagle medium (DMEM-F12) (Thermo Fisher Scientific, Courtaboeuf, France) and MKN74 (human intestinal type of GC, HSSR Bank JCRB0255) were cultivated in the Roswell park memorial institute (RPMI) 1640 Glutamax medium (Thermo Fisher Scientific), as previously described [9,25]. Both media were supplemented with 10% heat-inactivated fetal bovine serum (FBS, Thermo Fisher Scientific), and vancomycin at 50 μg/mL (Invitrogen, Waltham, MA, USA) in a humidified environment with 5% CO_2_ at 37 °C.

### 2.2. CRISPR-cas9 Mediated Nrf2 Knock-Out

Two sgRNA target DNA sequences (NRF2#1: 5′-ACTGGGCTCTCGATGTGACC-3′; NRF2#2: 5′-GCGACGGAAAGAGTATGAGC-3′) respectively located in exon 3 and exon 1 of NRF2 gene were designed using the CRISPOR algorithm (crispor.tefor.net [26]). Alt-R^®^-crRNA corresponding to target sequences was purchased from Integrated DNA Technologies (IDT) as well as human crRNA negative control and resuspended to 100 µM in Tris-EDTA (IDT). Alt-R^®^ S.p-Cas9HIFIv3 (4.5 pmol corresponding to 750 ng, IDT) was mixed with a 1.2 excess molar ratio of two-part gRNA (Alt-R^®^-crRNA + Alt-R^®^-tracrRNA) reconstituted following the supplier’s recommendations (IDT). Forward transfection was performed on 10,000 cells plated the day before using Lipofectamine CRISPRMAX reagent (Thermo Fischer Scientific, Waltham, MA, USA). After 10 min of incubation at RT, the solution was added in a 96 well containing 10,000 cells plated the day before in a 100 µL culture medium. At 2–3 days post-transfection, cells were trypsinized and half of them were cultured while the other half was pelleted, lysed and used as PCR template using Phire Tissue Direct PCR Master Mix (ThermoFisher Scientific). The PCR amplification of the targeted area was conducted following supplier instruction with primers 5′-CAAACGGGGTCATGACTGGT-3′ and 5′-TCCCAGATCAGACGTCAGGT-3′ for exon 3 and with primers 5′-CCCACTTCCCACCATCAACA-3′ and 5′-ACCTGGGAGTAGTTGGCAGA-3′ for exon 1. Sequencing of PCR products was performed by Eurofins Genomics and Sanger data were used to quantify Indels reflecting gene KO with DECODR algorithm (decodr.com [27] accessed on 19 November 2019) or ICE algorithm (ice.synthego.com [28] accessed on 19 November 2019). Cells were FACS-sorted to seed 1 cell/well in 96-well plates, and we let them expand to obtain cellular clones, which were then characterized by Sanger sequencing and Western blot analysis.

### 2.3. H. pylori Strains Culture and Co-Culture Model

Several *H. pylori* strains were used: the P12 WT strain and its isogenic mutants deleted for vacA (ΔvacA), cagA (ΔcagA) or cagE (ΔcagE), an essential protein for the assembly and function of the T4SS (from Rainer Haas, Ludwig Maximilian University of Munich, Munich, Germany). *H. pylori* were cultivated on Wilkins Chalgren agar plates supplemented with 10% human blood under microaerophilic (5% O_2_) conditions at 37 °C in a humidified atmosphere, as previously described [9]. Co-culture experiments were performed at a multiplicity of infection of 25:1 for the HFE-145 and AGS cells and 50:1 for the MKN74 cells. Evaluation of the bacterial number was estimated by spectrophotometry with OD_600 nm_ = 1 corresponding to 2 × 10^8^ bacteria/mL. Phase-contrast microscopy images of cell cultures were taken using an inverted phase-contrast Zeiss microscope equipped with a ×20 objective and a Zeiss acquisition software (Munich, Germany). The number of mesenchymal or vacuolated cells were manually counted over the total number of cells per picture. The percentage of positive cells was determined on at least 500 cells per condition.

### 2.4. ARE-Luciferase and GSH-GSSG Biochemiluminescent Assays

To measure Nrf2 activity, cells were seeded in 96-well plates at a density of 15,000 cells/well for 24 h and were then transfected with a control or a transfection-ready ARE luciferase reporter vector according to the manufacturer’s instructions (ARE Reporter Kit, BPS Bioscience, San Diego, CA, USA) 6 h before infection. Cells were lysed and the levels of ARE-luciferase activity were measured using the Dual-Luciferase^®^ reporter assay system (Promega, Madison, WI, USA) according to the manufacturer’s instructions. The level of chemiluminescence reflects the Nrf2 transcriptional activity. To evaluate the glutathione (GSH) level, cells were seeded in 96-well plates at a density of 5000 cells/well for 24 h prior to infection and the level of total GSH or oxidized GSH (GSSG) was evaluated using the GSH/GSSG-Glo^TM^ assay (Promega) according to the manufacturer’s instructions. The chemiluminescence measured allowed one to determine the GSSG/GSH ratio, where an increase indicates an increased oxidative stress. For both assays, the luciferase activity was measured using a Lumat LB850 (Berthold, Bad Wildbad, Germany).

### 2.5. Cellular ROS Level

Cells were seeded in 96-well plates at a density of 5000 cells/well for 24 h prior to infection. At the designated time, the cells were washed with PBS and stained with 5 µmol/L CellRox^TM^ green (Thermo Fisher Scientific) in PBS for 30 min at 37 °C. Cells were then fixed with 4% paraformaldehyde solution in cytoskeletal buffer and then counterstained with 50 mg/mL 4′-6-diamino-phenyl-indol (DAPI) (all from Thermo Fisher Scientific). Fluorescence intensity was measured using a CLARIOstar plate reader (BMG Labtech, Ortenberg, Germany).

### 2.6. Protein Extraction and Western Blot

Cells were plated in 6-well plates at a density of 200,000 cells/well for 24 h prior to infection. Cells were then washed twice with ice-cold PBS and lysed with Laemmli lysis buffer containing 8% sodium dodecyl sulfate (SDS), 40% glycerol, 0.01% bromophenol blue, 12% β-mercaptoethanol, 200 mM Tris at pH 6.8 (All from Thermo Fisher Scientific). The Western blot was performed, as previously described [29]. The primary antibodies used were anti-Nrf2 (1:2000, Abcam, Cambridge, UK), mouse anti–α-tubulin (1:2500, Sigma-Aldrich, St. Quentin Fallavier, France) and the secondary were Starbright Blue700 goat anti-mouse and anti-rabbit fluorescence conjugated secondary antibodies (1:2500, Bio-Rad, Marnes-la-Coquette, France). Immunoblotting band intensity was measured using the Fiji software (National Institutes of Health, Bethesda, MD, USA [30]).

### 2.7. RNA Extraction and RT-qPCR

Cells were plated in 12-well plates at a density of 100,000 cells/well for 24 h prior to infection. Total RNA was extracted using TRIzolTM reagent (Thermo Fisher Scientific) according to the manufacturer’s recommendations. Reverse transcription and real-time PCR were performed, as previously described [29]. The primer sequences are presented in Table 1. Analysis of amplified RNA samples was carried out using the 2^−ΔΔCT^ method with TBP and HPRT1 as reference genes.

### 2.8. Immunofluorescence

Cells were seeded on glass coverslips in 24-well plates at a density of 50,000 cells/well for 24 h prior to infection. Immunofluorescence staining was performed as previously described [11,29]. Primary antibodies used were rabbit anti-ZEB1 (1:100, Bethyl Laboratories, Souffelweyersheim, France A301-922) and rabbit anti-SNAIL (1:50, StemCell Technology, Vancouver, Canada 28193HCI30). The secondary antibody used was goat anti-rabbit Alexa-488-labelled (1:250, Thermo Fisher Scientific A32731). Images were taken using ×40 (numerical aperture, 1.3) oil immersion objective on an Eclipse 50i epi-fluorescence microscope (Nikon, Minato-ku, Tokyo, Japan) with Nis Element acquisition software. Fiji software was used for the relative quantification of the signal intensity on digital images [30].

### 2.9. Differential Proteomic Analysis

Cells were plated in 6-well plates at a density of 200,000 cells/well for 24 h and total proteins (three replicates for each cell lines) were extracted using Mem-PER^TM^ Plus Membrane Protein Extraction Kit (Thermo Fisher Scientific) according to the manufacturer’s recommendations. The entire protocol is described in the Appendix A. Briefly, protein samples were prepared for protein digestion and submitted to a nLC(C18)-MS/MS analysis (Orbitrap Fusion Lumos, Thermo Fisher Scientific) and Label-Free Quantitative Data Analysis. After validation and normalization, protein ratios between Nrf2-KO and WT for each cell line were calculated. Proteins with an abundance ratio >1.5 or <0.65 and an adjacent *p*-value < 0.05 were considered respectively to be significantly upregulated or downregulated in Nrf2-KO compared to WT cells. Prior to bioinformatics analysis, the resulting protein list was filtered to eliminate common contaminants. Protein network analyses were conducted using the STRING database (http://string-db.org accessed on 18 January 2021). All networks were visualized with Cytoscape (http://www.cytoscape.org/ accessed on 18 January 2021). Categorical annotations of proteins were supplied from Uniprot (https://www.uniprot.org/ accessed on 18 January 2021).

### 2.10. Immunohistochemistry on Human Gastric Biopsies

Ethic statements on human tissue samples were described in [9,11]. Tissue sections of 3 µm thick were prepared from formalin-fixed paraffin-embedded human tissues and immunostaining was performed with rabbit anti-*Helicobacter pylori* (1:300, 1 h, Agilent Technologies, Santa Clara, CA, USA B0471), rabbit anti-Nrf2 (1:1000, 1 h, Abcam ab137550), rabbit anti-ZEB1 (1:100, 2 h, Bethyl Laboratories A301-922A), mouse anti-Ki67 (1:75, 30 min, Agilent Technologies M7240 clone MIB-1) antibodies. Horseradish peroxidase-labeled secondary antibodies used were ImmPRESS^®^ HRP Horse Anti-Rabbit IgG Polymer Detection Kit (*Helicobacter pylori* and Ki67) or VECTASTAIN^®^ Elite ABC-HRP Kit, Peroxidase, R.T.U. (Universal) (Nrf2 and ZEB1) (Vector Laboratories, Newark, CA, USA) during 30 min. Liquid diaminobenzidine-chromogen substrate (Agilent Technologies) was used to visualize the staining and Mayer′s hemalum solution (Sigma Aldrich Chimie, Saint Quentin Fallavier, France) to counterstain. Slides were mounted with Mounting media, Eukitt^®^ (ORSAtec GMBH, Bobingen, Germany).

For all markers, more than 600 epithelial cells were counted, and the staining was assessed separately in two zones: (i) in the gastric pits and the isthmus and (ii) in the neck and base. For Ki67 evaluation, the percentage (%) of positive epithelial cells in body-type gastric mucosa was assessed. For ZEB1 evaluation, a mixed score was employed, including labeling (0% = 0; 1–25% = 1; 26–50% = 2; 51–75% = 3; 76–100% = 4) and staining (weak = 1; moderate = 2; intense = 3) indexes as previously described [31]. For Nrf2 evaluation, the percentage of moderate and intense nuclear immunoreactivity in epithelial cells was assessed. *Helicobacter pylori* immunoreactivity was assessed as follows: negative = 0; positive = 1 (focused immunopositivity); and 2 (diffused immunopositivity).

### 2.11. In Silico Database Analysis

UALCAN cancer database (http://ualcan.path.uab.edu/ accessed on 28 June 2022) [32] was used to investigate gene expression levels of NRF2 (gene NFE2L2) in normal gastric tissues and *H. pylori*-infected gastric tumors. Overall survival probability of GC patients was analyzed using the KMplot database tool (www.kmplot.com accessed on 28 June 2022) [33]. The JetSet probes used were 239952_at (ZEB1) and 201146_at (NFE2L2, NRF2). Analysis was conducted on all GC cases, or according to their Lauren classification status: diffuse type and intestinal type GC. High or low expression levels in patients’ samples were determined according to the software’s best cut-off value auto-setting and *p*-values were calculated by a log-rank test.

### 2.12. Statistics

Results are expressed as mean ± S.E.M of at least three independent experiments. Statistical tests were carried out using the GraphPad Prism software version 8.0.2. The Student’s *t*-test was used for the two-groups comparisons and one-way ANOVA with Tukey’s or Dunnet’s test as a post hoc test was performed for multiple comparisons.

## 3. Results

### 3.1. Nrf2 Signaling Pathway Is Modulated upon Infection with Helicobacter pylori

#### 3.1.1. *H. pylori* Infection Kinetics Is Associated with a Decreased Nrf2 Activity Consistent with Increased Oxidative Stress at 24 h Post Infection

To evaluate the modulation of Nrf2 upon infection, three gastric epithelial cell lines (HFE-145: non-cancerous, AGS: diffuse type GC and MKN74: intestinal type GC) were infected with *H. pylori* P12 strain for 1, 5 or 24 h. We observed a significant increase in Nrf2 activity measured by the ARE transcriptional activity 1 h post-infection (hpi) (Figure 1A). However, 24 hpi, the activity of Nrf2 was normalized to basal levels in the non-cancerous cell line and significantly inhibited in the cancerous ones (Figure 1A). This was associated with a significant decrease in Nrf2 protein level (Figure 1B,C). The mRNA of Nrf2 and two of its target genes, HO1 and NQO1, were more expressed in both GC cell lines than in the non-cancerous one, but the kinetic trend did not follow Nrf2 activity and protein level, suggesting post-transcriptional regulations (Figure 1D).

Infection with *H. pylori* increased the ROS production at 24 hpi, more particularly in both cancerous cell lines (Figure 2A,B). This might suggest that non-cancerous cells are able to handle the ROS accumulation in a more efficient way than the cancerous ones. Moreover, we observed an increased ratio of oxidized glutathione (GSSG) over total glutathione (GSH) (Figure 2C) at the same timepoint and in the three cell lines. This increased oxidative stress could be explained by the major drop in glutathione level 24 hpi observed in the three cell lines (Figure 2D). We observed that the non-cancerous cell line has a lower basal level of ROS and GSH compared to the two cancerous cell lines which seems to indicate a deregulation of GSH level and ROS elimination in gastric cancerous cells.

#### 3.1.2. *H. pylori* Virulence Factor VacA Is Involved in the Regulation of Nrf2 Activity

To decipher which *H. pylori* virulence factor among VacA, CagA or the T4SS is involved in the Nrf2 activity modulation, we infected the three cell lines with the P12-WT strain, VacA-deleted strain (ΔVacA), CagA-deleted strain (ΔCagA) or a T4SS-deficient strain (ΔCagE) for 24 h. We first validated our mutant *H. pylori* strains with phenotypic evaluation. As expected, infection with the WT strain was associated with an increase in both mesenchymal and vacuolated cells induced by CagA and VacA, respectively. Infection with ΔVacA was associated only with an increase in mesenchymal cells, also called hummingbird phenotype [34], and infection with ΔCagA or ΔCagE was associated only with an increase in vacuolated cells compared to non-infected cells (Figure 3A–C). An analysis of ARE transcriptional activity after infection with the mutant strains revealed some discrepancies depending on the cell line. In the absence of active VacA, Nrf2 transcriptional activity was restored compared to the non-infected in all three cell lines, indicating an implication of this virulence factor in the decrease in Nrf2 activity (Figure 3D). However, we cannot exclude a potential implication of CagA in the transcriptional activity of Nrf2 in the non-cancerous cells as well as an implication of the T4SS (observed with the ΔCagE strain) in the non-cancerous and diffuse type GC cells, even though the effect is less significant than with the ΔVacA strain and not observable in all the three cell lines (Figure 3D).

### 3.2. Nrf2 Impairment Favors H. pylori-Induced Epithelial-to-Mesenchymal Transition

Our team has previously demonstrated that *H. pylori* infection leads to an EMT-like transition favoring gastric carcinogenesis [9]. To study the role of Nrf2 in *H. pylori*-induced EMT, we knocked-out Nrf2 in the three studied gastric cell lines by CRISPR-cas9 (Appendix A). The modification of the proteomic network in the Nrf2-KO cells is shown in Appendix A. We included proteins whose differential expression demonstrated statistically significant changes by at least a fold of 1.5 or above for the upregulated or less than 0.65 for the downregulated proteins. Nrf2-KO caused the downregulation of 94 proteins and the upregulation of 70 proteins in non-cancerous cells. In the diffuse type GC cells, it induced the downregulation of 107 proteins and the upregulation of 101 proteins, while in the intestinal type GC, it induced the downregulation of 171 proteins and the upregulation of 113 proteins (Appendix A). Among the proteins that were down-regulated in Nrf2-KO cells, biological processes were logically mainly related to cell redox homeostasis more particularly in intestinal type GC (Appendix A, blue nodes). Moreover, biological processes associated with the differentially up-regulated proteins were mostly cell cycle-related events, notably mitotic regulation (Appendix A, pink nodes).

#### 3.2.1. Knock-Out of Nrf2 Favors Decrease in Epithelial Proteins Expression and Increase in Mesenchymal Ones

In our proteomic analysis dataset, we specifically looked for proteins described as epithelial (Figure 4A) and mesenchymal (Figure 4B) markers in the EMTome database [35]. Comparing Nrf2-KO vs. WT cells, we noted that different proteins were impacted by the KO of Nrf2 in the non-cancerous or the cancerous cell lines. We observed that many epithelial markers were under-expressed while many mesenchymal markers were overexpressed in Nrf2-KO cells for the three cell lines and especially in the non-cancerous and intestinal type well-differentiated cell lines compared to the poorly differentiated diffuse type. Therefore, the invalidation of Nrf2 leads to a decrease in epithelial markers and an increase in mesenchymal ones, suggesting a role of Nrf2 in maintaining an epithelial phenotype.

#### 3.2.2. Knock-Out of Nrf2 Favors *H. pylori*-Induced Mesenchymal Phenotype Acquisition and EMT Transcription Factor Nuclear Expression

We then evaluated the impact of Nrf2-KO in the context of *H. pylori* infection. Consistent with the results obtained in the proteomic analysis, we observed an increase in the percentage of cells harboring a mesenchymal phenotype in all three Nrf2-KO cell lines compared to WT (Figure 5A). Moreover, in the cancerous cell lines infected with *H. pylori*, we observed an increase in cells harboring a mesenchymal phenotype in Nrf2-KO cells compared to WT cells, significant for the intestinal GC (Figure 5A) suggesting that Nrf2 invalidation favors *H. pylori*-induced EMT. To confirm the results obtained with phenotypic analysis, we analyzed the nuclear expression of two EMT transcription factors, ZEB1 and SNAIL, reflecting their activity. We noticed that Nrf2-KO cells infected with *H. pylori* expressed more nuclear ZEB1 for the non-cancerous cells and more nuclear SNAIL for the three cell lines (Figure 5B,C).

### 3.3. H. pylori-Infected Patients Have Low Expression of NRF2 Which Is Inversely Correlated to ZEB1 Expression and Related to Poor Overall Survival Probability

We assessed the level of Nrf2 in gastric mucosa tissue sample of patients infected or not by *H. pylori*. First, the presence of the bacterium was detected in all infected tissues as well as some intestinal metaplasia areas characteristic of *H. pylori* infection (Appendix A). As already published by our team, infection with *H. pylori* was associated with a significant increase in ZEB1 and Ki67 proliferation marker in human gastric mucosa [11] (Figure 6A–C). When assessing the expression of Nrf2 in the entire gastric glands, we could not detect a difference in the percentage of nuclear Nrf2 positive cells between *H. pylori* negative or positive tissues. However, by separating the glands into two parts with the top composed of pit and isthmus and the bottom composed of neck and base, we observed an almost significant (*p* = 0.06) decrease in nuclear positive cells in the neck and base of *H. pylori* positive tissue compared to non-infected tissue (Figure 6B). To corroborate our results in another patient cohort, we checked the expression of NRF2 using the UALCAN database, which confirmed that NRF2 expression is decreased in *H. pylori*-infected gastric mucosa (Figure 7A). However, ZEB1 expression is not significant different between normal or *H. pylori*-infected gastric mucosa (Figure 7B).

Given that gastric mucosa histology differs by anatomical region, we assessed the status of Nrf2, ZEB1 and Ki67 in gastric epithelial cells of the two histologically and functionally different regions: (i) fundic or body type comprising the largest part of the stomach and containing fundic glands secreting gastric acid and (ii) pyloric type, which contains antro-pyloric glands that secrete alkaline mucous. In a qualitative analysis of consecutive serial sections, we noted that *H. pylori*-infected cases revealed ZEB1 immunopositivity in proliferating Ki67 epithelial cells along with limited Nrf2 expression (Figure 6D) suggesting a low expression of Nrf2 in cells going through EMT. Occasionally, in the same areas, ZEB1 and Nrf2 staining had a mutually exclusive pattern. In *H. pylori* negative cases, both ZEB1 and Nrf2 expression was limited in proliferating cells (Figure 6D).

Finally, we checked the overall survival of GC patients using the KMplotter™ analysis [27] according to the subtypes of GC. High expression of NRF2 is associated with a better survival, significant for all subtypes together and for the intestinal type of GC (Figure 7C). On the contrary, high expression of ZEB1 is associated with a poorer survival for all subtypes together and for the diffuse and intestinal subtypes separately (Figure 7D). Moreover, the analysis according to the expression ratio of NRF2 and ZEB1 (NRF2/ZEB1) revealed that patients having higher mean expressions of NRF2 compared to ZEB1 (NRF2/ZEB1 > 1) have better prognosis outcomes than those with lower NRF2 than ZEB1 (NRF2/ZEB1 ≤ 1), significant whatever is the GC subtype (Figure 7E). This suggests that patients with GC cells undergoing EMT and having less NRF2, similar to what is described in the histological analysis (Figure 6), have a lower survival probability.

## 4. Discussion

In this study, we aimed at assessing the regulation of the Nrf2 expression and activity upon *H. pylori* infection in gastric epithelial cells. We hereby demonstrated that infection with a CagA+VacA+ *H. pylori* strain decreases Nrf2 activity and protein level 24 h after infection. Nrf2 seems to be a guardian of the epithelial integrity, as its invalidation leaded to an increased EMT program even exacerbated in the context of *H. pylori* infection. Consistently, Nrf2 was decreased in *H. pylori*-infected gastric mucosa while ZEB1, one of the main EMT-TF, was increased. We therefore propose the hypothesis that Nrf2 downregulation upon infection may participate in the EMT induced by *H. pylori*, an important feature of gastric carcinogenesis.

One strength of our study was the use of three different gastric epithelial cell lines allowing the study of a non-cancerous model (HFE-145) and two different cancerous models representing the two most common subtypes of GC: diffuse (AGS) and intestinal (MKN74). As we could expect from the literature [36,37,38], cancerous cells expressed more NRF2, HO1 and NQO1 mRNA, more ROS and more GSH at the basal level compared to non-cancerous ones, validating the used cellular models (Figure 1 and Figure 2). We also observed differences between the two subtypes of GC. Intestinal type GC, characterized by malignant cohesive epithelial cells and intestinal-type glandular differentiation infiltrating the tissue, is the most common type, occurring in about 54% of cases, while diffuse type GC, found in about 32% of cases, contains poorly differentiated and poorly cohesive tumor cells [2]. Consistent with those differences, we observed more formation of mesenchymal-like cells upon infection of the diffuse than of the intestinal GC type and the non-cancerous gastric epithelial cells (Figure 3 and Figure 5).

Our results demonstrated that infection with *H. pylori* induces early activation of Nrf2 followed by a decrease in its activity and protein level (Figure 1). Few studies have investigated the modulation of Nrf2 upon *H. pylori* infection. Paik and colleagues have infected AGS cells with the ATCC 43504 *H. pylori* strain for 24 h and have demonstrated a time-dependent accumulation of Nrf2. Our opposing results suggest a difference depending on the *H. pylori* strain used. Here, we used a CagA+VacA+ P12 strain and demonstrated that VacA could be responsible for the decreased activity of Nrf2 in the three cell lines. A possible implication of CagA, more especially in the non-cancerous HFE-145 cells, could not be excluded (Figure 3). Interestingly, it has been demonstrated that there is an interplay between these two virulence factors. For example, CagA reduces the entry of VacA into the cells and downregulated its vacuolating effect. On the contrary, VacA downregulates CagA activity but promotes its accumulation [5,39]. Further studies are needed to decipher the mechanisms involved in the *H. pylori* VacA and/or CagA-induced modulation of Nrf2 as well as the significance of the interplay between both virulence factors in oxidative stress.

The downregulation of Nrf2 activity and protein level, 24 hpi, was associated with the increased oxidative stress demonstrated by ROS accumulation and an increase in GSSG/GSH ratio (Figure 2). Considering the impact of oxidative stress in GC [15], these results suggest that the downregulation of Nrf2 may be a key point in GC initiation and progression. Moreover, we observed that *H. pylori* infection depletes cellular glutathione (GSH), the most abundant antioxidant protein. This depletion has already been described by others on cellular models [40,41] as well as in patients [41,42,43]. It has also been demonstrated that the eradication of *H. pylori* restores GSH levels in patients’ antral mucosa [44]. This result is of importance, knowing that molecular changes in the GSH antioxidant system and disturbances in GSH homeostasis have been implicated in tumor initiation, progression and treatment response [38]. Depletion of GSH after *H. pylori* infection may thus participate in gastric carcinogenesis.

In this study, we demonstrated that Nrf2 is a central actor of epithelial integrity and that its invalidation favors the EMT program (Figure 4). The role of Nrf2 in EMT has raised researchers’ interest in the past years but studies demonstrate contradictory results. Some studies have demonstrated that Nrf2 activation is associated with EMT induction [21,45,46] while others, as our results suggest, demonstrate that Nrf2 impairment leads to the induction of the EMT program [47,48,49] or, in the same way, that Nrf2 activation is associated with EMT inhibition [50,51,52]. Interestingly, it has also been described that Nrf2 may act as a phenotypic stability factor in restricting complete EMT by maintaining the cells in a hybrid epithelial/mesenchymal state [23,53]. Here, we demonstrated that Nrf2 invalidation exacerbates *H. pylori*-induced EMT, more especially in the cancerous cell lines. Indeed, Nrf2 invalidation increased the percentage of cells with a mesenchymal phenotype as well as a nuclear accumulation of ZEB1 and SNAIL, two EMT transcription factors (Figure 5). In patient tissues, we confirmed the low level of Nrf2 in *H. pylori*-infected tissues and we demonstrated that ZEB1+ cells lacked Nrf2 (Figure 6). Our study is the first to link Nrf2 modulation to EMT in the context of *H. pylori* infection and suggest that *H. pylori*-induced downregulation of Nrf2 may participate in the initiation of the EMT program. Consistent with those results, we also demonstrated that high expression of Nrf2 is associated with a better patients’ overall survival and that patients with high ZEB1 but low Nrf2 have a poorer overall survival (Figure 7), supporting the inverse association between these two transcription factors.

As a result of these findings, the early stages of *H. pylori*-induced GC may be reduced using Nrf2-targeted strategies. Several studies have already demonstrated that the use of natural Nrf2 activators decreases *H. pylori*-induced gastritis and oxidative stress [54,55,56,57,58]; hence, it would be of interest to determine whether these compounds could prevent *H. pylori*-induced EMT. Importantly however, given the implication of Nrf2 in cancer progression [59,60,61], Nrf2-inducing strategies must be only employed during the early phase of the infection, such as gastritis, and not when GC is detected.

## 5. Conclusions

Infection with *Helicobacter pylori* decreases the activity and the protein level of Nrf2, a major transcription factor involved in the cellular antioxidant response. This decrease is associated with increased oxidative stress and a depletion of the glutathione level, the most abundant antioxidant protein. We demonstrated that Nrf2 is important to maintain the gastric epithelium integrity and that its *H. pylori*-induced downregulation participates in the induction of the EMT program. These results pave the way for new therapeutic strategies using Nrf2 modulators to combat oxidative stress and EMT induction by *H. pylori* infection during the early stage of gastric carcinogenesis.

## Figures and Tables

**Figure 1 cancers-14-04316-f001:**
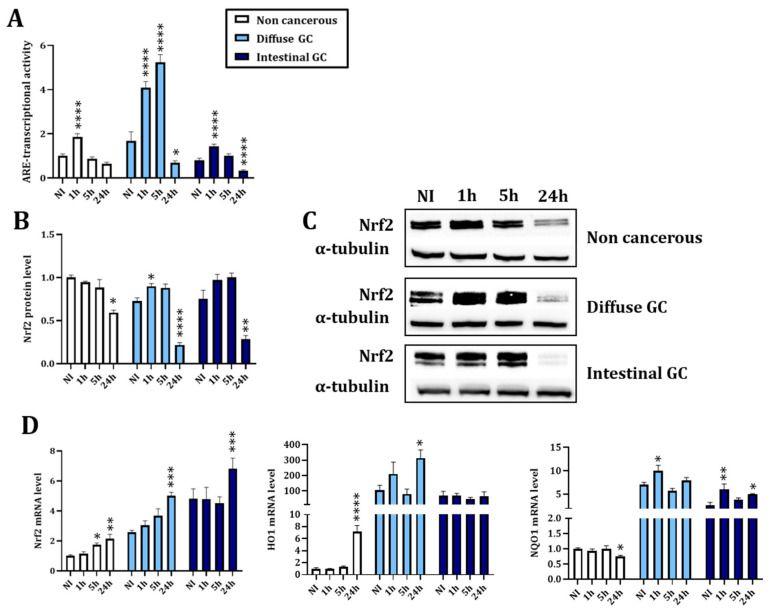
Kinetic infection with *H. pylori* modulates Nrf2expression and activity. (**A**) Evaluation of the Nrf2 activity by assessing the Antioxidant Responsive Element (ARE) transcriptional activity by a biochemiluminescent assay. (**B**) Evaluation of Nrf2 expression by Western blotting. The uncropped blots and molecular weight markers are shown in Appendix A. (**C**) Representative images of the Nrf2 Western blotting. (**D**) Evaluation of the mRNA level of Nrf2 and two of its common target genes by RT-qPCR. NI: uninfected cells. One hour, 5 h and 24 h are the timepoints of the infection kinetics. Mean ± SEM of 3–5 independent experiments. Statistics were performed per cell line with one-way ANOVA followed by a Tukey’s multiple comparisons test. All *p*-value < 0.05 were considered significant. *: significant difference compared to NI. * < 0.05, ** < 0.01, *** < 0.001 and **** < 0.0001.

**Figure 2 cancers-14-04316-f002:**
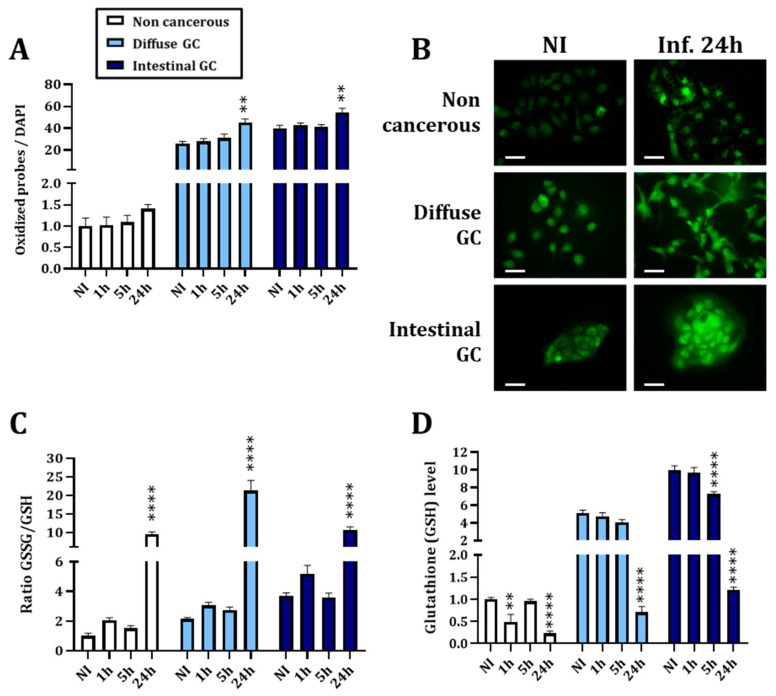
Infection with *H. pylori* increases oxidative stress and decreases the cellular glutathione level 24 h after infection. (**A**) Evaluation of ROS formation by assessing fluorescent probe signals normalized by the DAPI and read on a microplate reader in a 96-well plate. (**B**) The increase in fluorescent probes’ signal was also verified using an epifluorescence microscope. Scale bar: 20 µm. (**C**) Evaluation of the ratio between oxidized (GSSG) and total (GSH) glutathione, one of the major cellular antioxidant proteins, by a biochemiluminescent assay. An increase in this ratio depicts increased oxidative stress. (**D**) Evaluation of the level of cellular glutathione by a biochemiluminescent assay. NI: uninfected cells. One hour, 5 h and 24 h are the timepoints of the kinetics of infection. Mean ± SEM of 3–5 independent experiments. Statistics were performed per cell line with one-way ANOVA followed by a Tukey’s multiple comparisons test. All *p*-value < 0.05 were considered significant. *: significant difference compared to NI. ** < 0.01 and **** < 0.0001.

**Figure 3 cancers-14-04316-f003:**
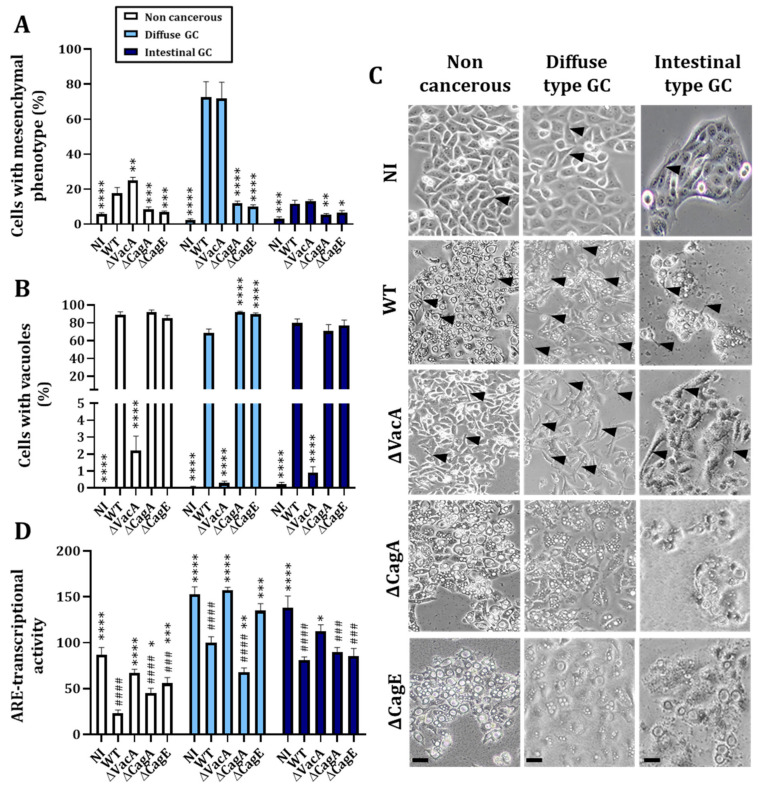
The virulence factor VacA is implicated in the decreased Nrf2 activity observed 24 h after infection with *H. pylori*. (**A**) Evaluation of the percentage of cells with a mesenchymal phenotype performed by phase contrast microscopy. (**B**) Evaluation of the percentage of cells with vacuoles performed by phase contrast microscopy. (**C**) Representative images of phase contrast microscopy of the three cell lines infected with the WT or with mutant *H. pylori* strains. Black arrows show cells with a mesenchymal phenotype. (**D**) Evaluation of the Nrf2 activity by assessing the Antioxidant Responsive Element (ARE) transcriptional activity by a biochemiluminescent assay. NI: uninfected cells. WT: *H. pylori* P12 wild type strain-infected cells. ΔVacA: *H. pylori* P12 VacA deleted strain-infected cells. ΔCagA: *H. pylori* P12 CagA deleted strain-infected cells. ΔCagE: *H. pylori* P12 CagE, an essential protein for the T4SS assembly, deleted strain-infected cells. Mean ± SEM of 3–4 independent experiments. Statistics were performed per cell line with ANOVA followed by a Dunnet’s multiple comparisons test to compare all conditions to NI (#) or to WT (*). All *p*-value < 0.05 were considered significant. * < 0.05, ** < 0.01, ^###/^*** < 0.001 and ^####/^**** < 0.0001.

**Figure 4 cancers-14-04316-f004:**
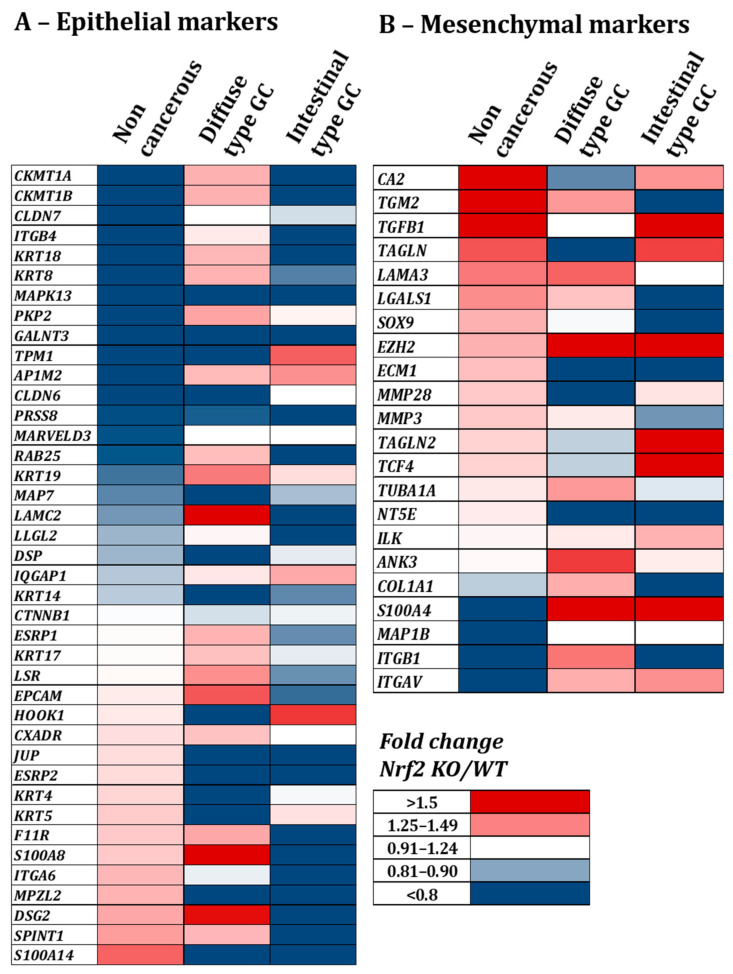
Nrf2 knock-out decreases epithelial and increases mesenchymal proteins in gastric non-cancerous, diffuse cancer or intestinal GC cell lines. Heat map from a proteomic analysis showing fold change between Nrf2-KO and Nrf2-WT for epithelial (**A**) and mesenchymal (**B**) proteins selected from the EMTome database [35]. Proteins were classified in increasing order of the fold change Nrf2-KO/Nrf2-WT of the non-cancerous cell line. *n* = 3 for each cell line.

**Figure 5 cancers-14-04316-f005:**
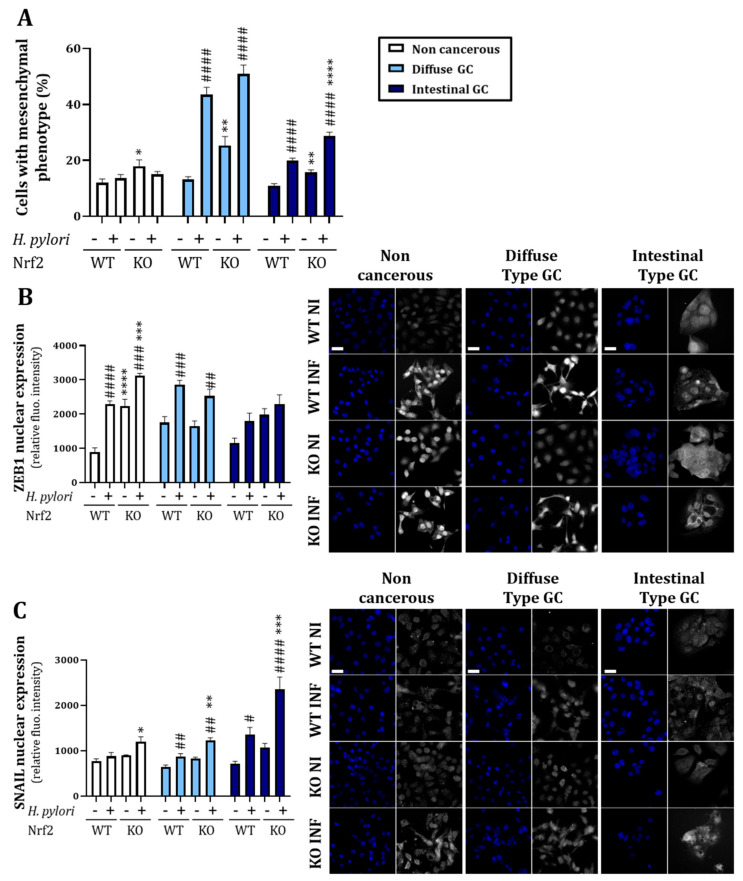
Acquisition of a mesenchymal phenotype associated with an increased ZEB1 and SNAIL nuclear expression is exacerbated in Nrf2-KO cells infected with *H. pylori* for 24 h. (**A**) Evaluation of the percentage of cells with a mesenchymal phenotype performed by phase contrast microscopy. (**B**) Evaluation of nuclear intensity of ZEB1 by immunofluorescence and representative images with DAPI in blue and ZEB1 in grey. Scale bar: 20 µm. (**C**) Evaluation of nuclear intensity of SNAIL by immunofluorescence and representative images with DAPI in blue and SNAIL in grey. Scale bar: 20 µm. WT: Nrf2-wild type. KO: Nrf2-knock out cells. NI: uninfected cells. INF: infected with *H. pylori*. Mean ± SEM of 3–5 independent experiments. Statistics were performed per cell line with ANOVA followed by a Tukey’s multiple comparisons test. All *p*-value < 0.05 were considered significant. *: effect of the KO when compared to the WT of the same condition (uninfected or uninfected). #: effect of the infection when compared to the uninfected of the same condition (WT or KO). *^/#^ < 0.05, **^/##^ < 0.01, ***^/###^ < 0.001 and ****^/####^ < 0.0001.

**Figure 6 cancers-14-04316-f006:**
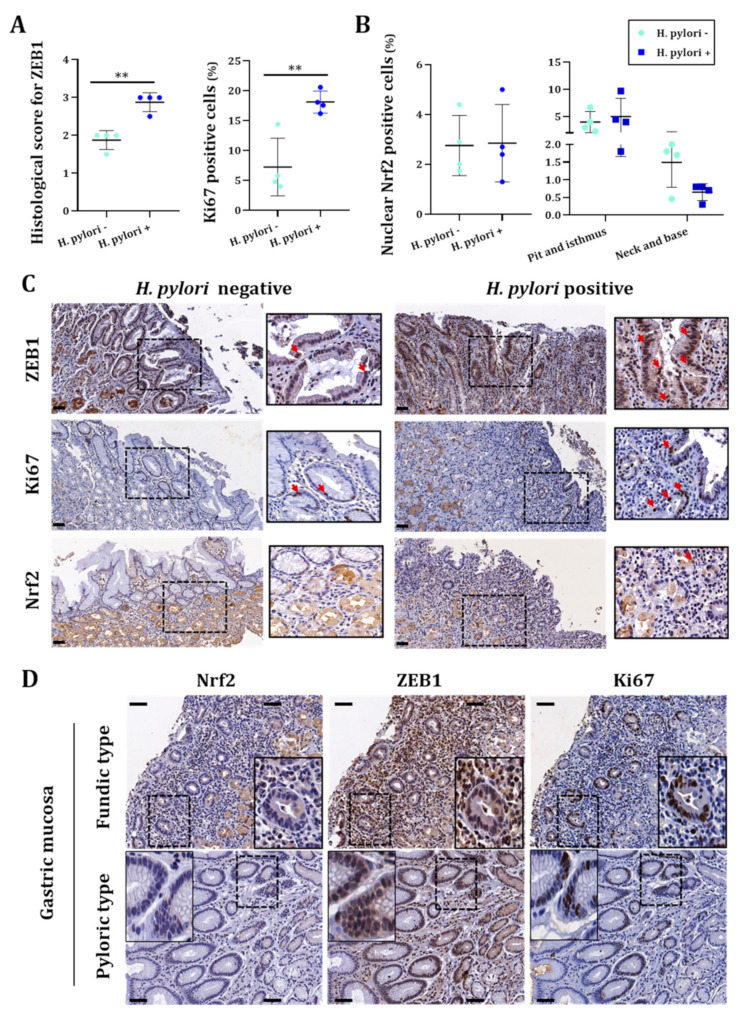
*H. pylori* infection tends to decrease Nrf2 expression in the bottom of gastric glands and is inversely correlated with the presence of ZEB1. (**A**) Evaluation of ZEB1 (histological score) and Ki67 (percentage of positive cells) in *H. pylori* negative or positive tissues by immunohistochemistry. Statistics were performed with a *t*-test. All *p*-value < 0.05 were considered significant. ** < 0.01. (**B**) Evaluation of Nrf2 (percentage of positive cells) in *H. pylori* negative or positive tissues by immunohistochemistry in the entire gland (left) or by separating the top (pit and isthmus) and the bottom (neck and base) of gastric glands (right). (**C**) Representative pictures of immunohistochemistry for ZEB1, Ki67 and Nrf2 expression in *H. pylori* negative or positive human gastric tissue samples. Red arrows denote immunopositivity. Scale bar: 50 µm. (**D**) Qualitative analysis of consecutive serial sections for Nrf2, Ki67 and ZEB1 expression in the gastric mucosa (fundic and antro-pyloric areas) of *H. pylori*-infected patients. Scale bar: 50 μm.

**Figure 7 cancers-14-04316-f007:**
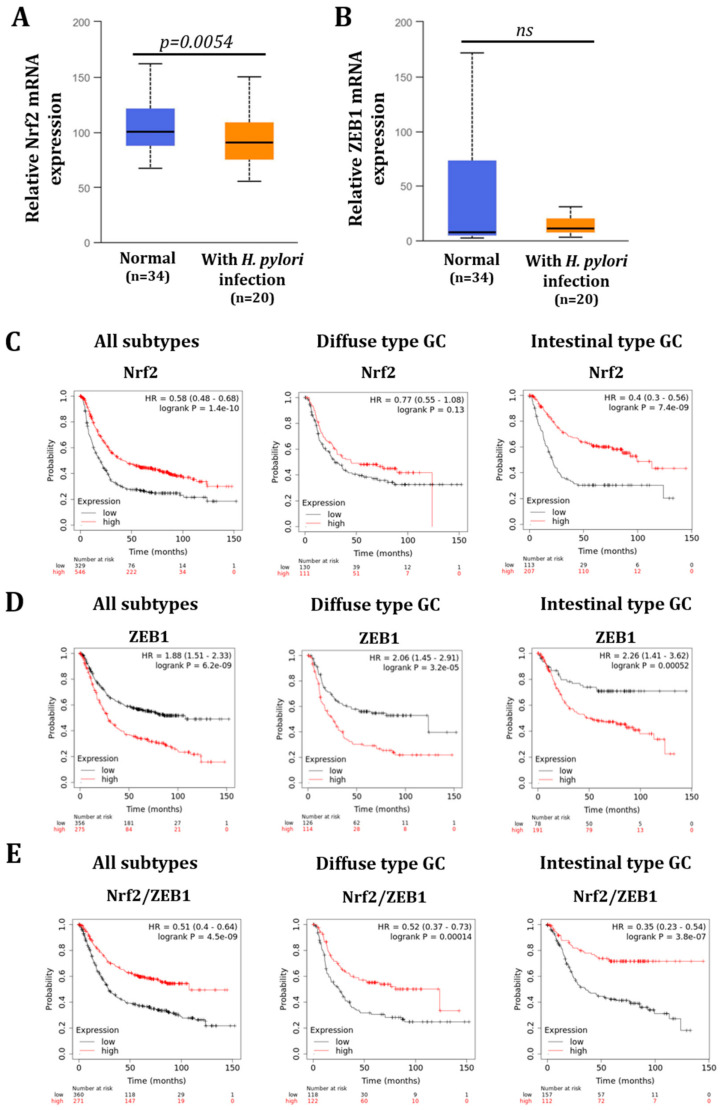
In silico analyses reveal that Nrf2 expression is down-regulated in patients’ *H. pylori*-infected gastric tissues and is associated to a poorer overall survival probability. (**A**,**B**) Evaluation of NRF2 (**A**) and ZEB1 (**B**) gene expression in normal or *H. pylori*-infected gastric tissues assessed using the UALCAN cancer database. (**C**,**D**) KMplot database analyses showing the overall survival probability of GC patients according to NRF2 (**C**) and ZEB1 (**D**) expression for all subtypes or diffuse type GC or intestinal type GC. (**E**) KMplot database analyses showing the overall survival probability of GC patients according to NRF2/ZEB1 expression ratio for all subtypes or diffuse type GC or intestinal type GC.

**Table 1 cancers-14-04316-t001:** Primer sequences used in the study.

Gene	Forward (5′-3′)	Reverse (5′-3′)
NFE2L2 (Nrf2)	CACATCCAGTCAGAAACCAGTGG	GGAATGTCTGCGCCAAAGCTG
HMOX1 (HO1)	CCAGGCAGAGAATGCTGAGTTC	AAGACTGGGCTCTCCTTGTTGC
NQO1	CCTGCCATTCTGAAAGGCTGGT	GTGGTGATGGAAAGCACTGCCT
HPRT1	TGGTCAGGCAGTATAATCCA	GGTCCTTTTCACCAGCAAGCT
TBP	TGCACAGGAGCCAAGAGTGAA	CACATCACAGCTCCCCACCA

## Data Availability

The data presented in this study are available on request from the corresponding author.

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
