# Peer review of "Nrf2 Downregulation Contributes to Epithelial-to-Mesenchymal Transition in Helicobacter pylori-Infected Cells"

_cancers, 2022, doi:10.3390/cancers14174316_

Round 1

Reviewer 1 Report

Bacon et al. demonstrated a study that investigated the role of Nrf2 in Epithelial-to-Mesenchymal Transition (EMT) in Helicobacter pylori-infected cells. Nrf2 has been widely studied in the context of antioxidant response and chemoresistance, and published studies have shown the role of Nrf2 in EMT (PMID: 31329868, PMID: 35480877, PMID: 27982105 et al.). The current manuscript's value may mainly be based on the study of Helicobacter pylori-infection-Nrf2 reactions. Some concerns should be answered:

1. The authors emphasized that kinetic infection with H. pylori modulates the Nrf2 pathway. However, the whole manuscript only studied Nrf2 and did not study any other players or downstream targets of the Nr2 pathway.

2. In figure 3, the authors showed the mesenchymal phenotype by phase contrast microscopy. Why did the authors not test EMT markers (for example, E Cadherin, Vimentin, SNAIL, or ZEB1) immunofluorescence staining and show the results in figure 3? Instead, the authors jumped to proteomic analysis in figure 4 and went back to check the famous EMT markers ZEB1 and SNAIL without showing and identifying any other new markers, which makes the figure 4 data not that significant for the whole manuscript.

3.Figure6, immunofluorescence staining is needed to test the Nrf2/ZEB1 expressions and locations in the same tissue slides.

4. Figure 7, what is the relative ZEB1 mRNA expression difference in normal vs. with H. pylori infection? What are the IKM-plotter results for the relationship between ZEB1 and different gastric cancer subtypes' prognosis? Will ZEB1 itself alone, not correlated with Nrf2, determine the gastric cancer patient prognosis outcome?

Small issues:

1. The formats of all the figures in the manuscript are messed up with the page line numbers, making the figures very hard to read.

 2. The authors did not describe the “biochemiluniescent assay” used in figures 1 and 2 in the method.

Author Response

We would like to thank the Reviewer for his/her time and useful comments to improve our manuscript. Please find enclosed the point-by-point response.

Reviewer 2 Report

Comments on Nrf2 downregulation contributes to epithelial-to-mesenchymal 2 transition in Helicobacter pylori-infected cells

The work done by Bacon et al. is well planned but there are still many major concerns with the manuscript which have been highlighted below. The comments need to be addressed and revised thoroughly for publishing in Cancers

Additionally, the result section in the is poorly written and disordered, where quite a few observations (other than the ones the authors want to highlight) are not discussed at all. So, I hope authors could work on this section to make it a reliable and meaningful study.

1.        There have been quite a few studies on the effect of Nrf2 on EMT transition in pathologies like pulmonary fibrosis and cancer. The novelty of this present study lies in the fact that they have used H. pylori infected gastric cells as the pathological condition. So, please briefly mention these already published studies in the introduction section and include the references.

2.        In line no.84- 85 of Introduction section, the authors states hydrogen peroxide and ROS as separate entities, while H2O2 is on the major ROS types. Authors should rewrite the sentence accordingly.

3.        In the materials and methods, while mentioning about the cell lines used in the study, their origin should be included. It should be stated that these are cell lines derived from humans.

4.        Also, the authors should elaborate why different multiplicity of infections were used for the HFE-145, AGS and MKN74 cells.

5.        In Figure 1; Figure 1A, how would the authors explain the significant increase in ARE activity in 1 h and 5h time points for all the cell types especially the AGS cell line? Since the inhibition of ARE activity starts to show up at 24 hours, authors should add a 48-hour time point to this study.

6.        For the Figure 1B and C, the representative western blot for Nrf2 expression shown (has a decrease in Nrf2 expression) in AGS for 5h timepoint doesn’t match the evaluated blot result in 1B where no such decrease is noted. Please explain the disparity and if 1B is okay change the Nrf2 blot for Diffuse GC.

7.        In figure2, authors say that “Infection with H. pylori increased oxidative stress as observed by an increased ROS production (Figure 2A-B) and an increased ratio of oxidized glutathione (GSSG) over total glutathione (GSH)”, however this statement is partially true because increased oxidative stress is observed in GC cells but not in non-cancerous cells that are H. pylori infected. Can the authors explain this observation? Does this mean that H. pylori does not induce oxidative stress in non-cancerous cells? However, the GSSG/GSH ratio alters after 24 hours of exposure in this cell type also, how are this possible? Explain and please change the above-mentioned statement and the first line of the figure2 legend accordingly.

8.        Also, with line of the previous comment, since oxidative stress detection is a very important part of this study, it is highly recommended to measure cellular ROS level by CellROX™ Green Reagent via flow cytometry apart from the traditional fluorescence microscopy. This measurement would be more precise since the oxidatively stressed and non-stressed cells will be reliably distinguished from dead cells by flow cytometry when CellROX™ Green is used together with the SYTOX™ Red Stain and this may help in addressing the issue stated in my previous comment.

9.        From the experiments done and shown in Figure3 A, B and C it is quite clear that cells with mesenchymal phenotype increases in VacA deleted strain compared to the CagA and CagE deleted strains, but the method for cell counting greatly lacks detailing and needs to be included in the Materials and Methods section.

10.     The part of the study shown in Figure 3D, elucidating the role of VacA in regulation of Nrf2, needs to be explained well. The way the authors wrote this part is very confusing and inconclusive. From the ARE activity, it looks like when the 3 cell types are infected with VacA deleted strain, ARE activity increases. But when infected with the CagE deleted strain also, we see the similar phenotype, which is nowhere mentioned in the manuscript. It looks like authors themselves are not very confident about the credibility of their observation. Therefore, a western blot image of Nrf2 should be included in this study to show direct evidence and conclude the role of this deleted virulence factors on Nrf2. The line 38 in abstract, subheading in line 343 and figure legend in 389 should be rewritten based on the observation.

11.     In line 475-477, the authors state that “Moreover, in the cancerous cell lines infected with H. pylori, we observed an increase of cells harboring a mesenchymal phenotype in Nrf2-KO cells compared to WT cells, significant for the intestinal GC (Figure 5A) suggesting that Nrf2 invalidation favors H. pylori-induced EMT.” which holds only partially true from their observation, as their results also show that in the cancerous cells without H. pylori infection, irrespective of WT or KO, has much more mesenchymal phenotype than if infected with H. pylori. Then how can they say that Nrf2 invalidation favor only H. pylori induced EMT? Negative H. pylori cells also shows more ZEB1, and SNAIL expressions compared H. pylori positive cells, how is this possible? Is H. pylori infection inhibiting mesenchymal transition? This is very perplexing and kind of contradicts their previous observations, rather hugely questioning the validity of this study.

12.     Moreover, results indicate that SNAIL expression in H. pylori infected Nrf2 KO cells was not significantly more compared to H. pylori infected WT cells? How will the authors explain this? Once again, if this holds true then the actual role of Nrf2 needs to be totally reinstated. Also, the figure numbers in section 3.2.2 mentioned in the manuscript while explaining the observations are wrong and needs to be corrected.

13.     The authors should briefly introduce the Pyloric and Fundic type in the writeup before introducing them suddenly in the figures. Other than labelling of the axis in Figure 6, it is nowhere mentioned in the manuscript or figure legend. Also, they should explain why they chose these 2 types? In line 540, the authors mention about the “In a qualitative analysis of consecutive serial sections” the Fundus and the pylorus are situated quite far away from each other, then how can the authors claim them to be consecutive sections? Also, they should mention the significance of performing the experiment shown in Figure 6D.

14.     The authors should provide summarized interpretation of proteomic analysis shown in the Figure 4, where the whole observation can be understood in a simpler representation.

15.     The catalog numbers of all the antibodies should be included in the manuscript.

Author Response

(The authors gave the same response as above.)

Round 2

Reviewer 1 Report

All the concerns have been addressed.

Reviewer 2 Report

The authors should request more time and complete the suggested experiments for scientific enrichment of the manuscript.